# THOUGHT-RETRIEVER: DON'T JUST RETRIEVE RAW DATA, RETRIEVE THOUGHTS

## ABSTRACT

Large language models (LLMs) have transformed AI research thanks to their powerful *internal* capabilities and knowledge. However, existing LLMs still fail to effectively incorporate the massive *external* knowledge when interacting with the world. Although retrieval-augmented LLMs are proposed to mitigate the issue, they are still fundamentally constrained by the context length of LLMs, as they can only retrieve top-K raw data chunks from the external knowledge base which often consists of millions of data chunks. Here we propose *Thought-Retriever*, a novel model-agnostic algorithm that helps LLMs generate output conditioned on arbitrarily long external data, without being constrained by the context length or number of retrieved data chunks. Our key insight is to let an LLM fully leverage its intermediate responses generated when solving past user queries (**thoughts**), filtering meaningless and redundant thoughts, organizing them in thought memory, and retrieving the relevant thoughts when addressing new queries. Besides algorithmic innovation, we further meticulously prepare a novel benchmark, AcademicEval, which requires an LLM to faithfully leverage ultra-long context to answer queries based on real-world academic papers. Extensive experiments on AcademicEval and two other public datasets validate that Thought-Retriever remarkably outperforms state-of-the-art baselines, achieving an average increase of at least 7.6% in F1 score and 16% in win rate across various tasks. More importantly, we further demonstrate two exciting findings: (1) Thought-Retriever can indeed help LLM self-evolve after solving more user queries; (2) Thought-Retriever learns to leverage deeper thoughts to answer more abstract user queries.

## 1 INTRODUCTION AND RELATED WORK

Large language models (LLMs) have revolutionized AI research thanks to their powerful *internal* capabilities (Zhao et al., 2023; Wang et al., 2023) and knowledge (Peng et al., 2023b). When building LLMs, researchers further expect LLMs to interact with the world by effectively incorporating the *external knowledge* as their long-term memories, *e.g.*, collected from *facts* (Sun et al., 2023) or interactions with *other AIs* (Wu et al., 2023; Kannan et al., 2023). Importantly, the scale of the external knowledge for LLMs could be arbitrarily large; ultimately, all the digitized information within our universe could serve as the external knowledge for these LLMs. In practice, when building personalized LLM applications (Bill and Eriksson, 2023) or LLM-powered domain experts (Thirunavukarasu et al., 2023; Liu et al., 2023), *e.g.*, AI doctor, the relevant external knowledge for the LLMs could also easily get extremely large, *e.g.*, billions of tokens. Therefore, our paper aims to raise attention to the pressing research question: *how to effectively and efficiently help LLMs utilize (arbitrarily) rich external knowledge*.

To help LLMs better incorporate external knowledge, existing research mainly falls into two categories: *long-context LLMs* and *retrieval-augmented LLMs (RALMs)*. (1) *Long-context LLMs*, such as MPT (MosaicML, 2023) and LongChat (LM-SYS, 2023), aim to expand the LLM's context window, *e.g.*, via novel training algorithms (Tay et al., 2022), inference algorithms (Xiao et al., 2023), new architectures (Peng et al., 2023a; Gu and Dao, 2023), or system optimization (Xu et al., 2023). Although these methods improve the working memory size of LLMs, they cannot fundamentally address the issue of interacting with ultra-rich external knowledge using LLMs, since the computational complexity is often quadratic to the context length. (2) *RALMs* retrieve pertinent

Figure 1: **Why Thought-Retriever helps**. **(a)** A standard RALM is limited by the number of retrieved chunks. The retrieved data fails to cover all the necessary data chunks (red chunks) for a user query. **(b)** A hierarchical RALM retrieves summaries $S_i$, generated independently from user queries, which could improve recall at the cost of lower precision. **(c)** Thought-Retriever leverages past LLM thoughts collected from answering user queries, with little computational overhead. Thought-Retriever balances low-level facts and high-level thoughts, leading to high precision and recall.

information from external knowledge bases using retrievers, such as BM-25 (Robertson et al., 2009), Contriever (Izacard et al., 2022), and DRAGON (Lin et al., 2023). However, these algorithms are still constrained by LLMs' context length, since they can only retrieve top-K raw data chunks from the external knowledge that fits within an LLM's context limit. (3) *Hierarchical RALMs, e.g.,* creating a tree-structured memory for an LLM (Chen et al., 2023). Despite its potential to help LLMs incorporate more abstract knowledge, manually summarizing closed chunks and rigidly forming a tree structure proves to be a costly and inefficient method. This approach demands significant resources and lacks the flexibility to adapt to specific inputs in LLMS. Overall, existing methods in attempting to include external knowledge for LLMs still exhibit *fundamental limitations in efficiency and effectiveness*.

Here, we propose *Thought-Retriever*, an LLM-agnostic self-evolving retrieval framework that leverages historical LLM responses to answer new queries. Our key insight is that LLM responses can be transformed into *thoughts* with little computational overhead and that the thoughts can be organized as a thought memory for the LLM to facilitate future tasks. Psychological studies (Kurzweil, 2013; Snell, 2012) support our insight, revealing that human memory is organized hierarchically, which not only aids in retrieving relevant information for problem-solving but also gradually deepens our understanding of the world through continuous processing and summarizing these interactions into complex cognitive thoughts. Notably, through continuous interaction with diverse user queries, Thought-Retriever progressively generates more novel and expansive thoughts. This is achieved by organizing new data chunks from external knowledge into thoughts after addressing each query, filtering out meaningless and redundant thoughts, and ultimately incorporating high-quality thoughts into the thought memory. Therefore, Thought-Retriever gives an LLM the potential to *utilize arbitrarily rich external knowledge long-term memories and achieve self-evolution in capabilities*.

In addition to algorithmic advancements, we also meticulously developed a novel benchmark, *AcademicEval*, which challenges a Large Language Model (LLM) to accurately utilize extensive context to answer queries based on real-world academic papers. Our comprehensive experiments on AcademicEval and two additional datasets confirm that Thought-Retriever significantly surpasses state-of-the-art baselines, achieving an average increase of at least 7.6% in F1 score and 16% in win rate across various tasks. Furthermore, we present two intriguing discoveries: (1) Thought-Retriever can indeed facilitate the self-evolution of an LLM after addressing more user queries; (2) Thought-Retriever is capable of harnessing deeper insights to respond to more abstract user queries.

In summary, our *main contributions* are as follows: **(1)** Thought-Retriever framework enables an LLM to efficiently and effectively utilize external knowledge, and further allowing it to self-evolve through continuous interactions. **(2)** AcademicEval, a real-world benchmark for testing LLM's understanding of ultra-long context. **(3)** Thought-Retriever consistently outperforms all state-of-the-art retrieval-augmented and long-context baselines and presents exciting new findings.

## 2 THOUGHT-RETRIEVER: EFFECTIVELY EQUIP LLMs WITH EXTERNAL KNOWLEDGE

### 2.1 PRELIMINARIES

An *external knowledge* base $\mathcal{K} = (K_1, K_2, ..., K_n)$ consists of $n$ data chunks. An LLM $L$ can generate a *thought* $T_i = L(Q_{\text{think}}, \mathcal{K}_i)$ as its response when it is prompted to elaborate its thought

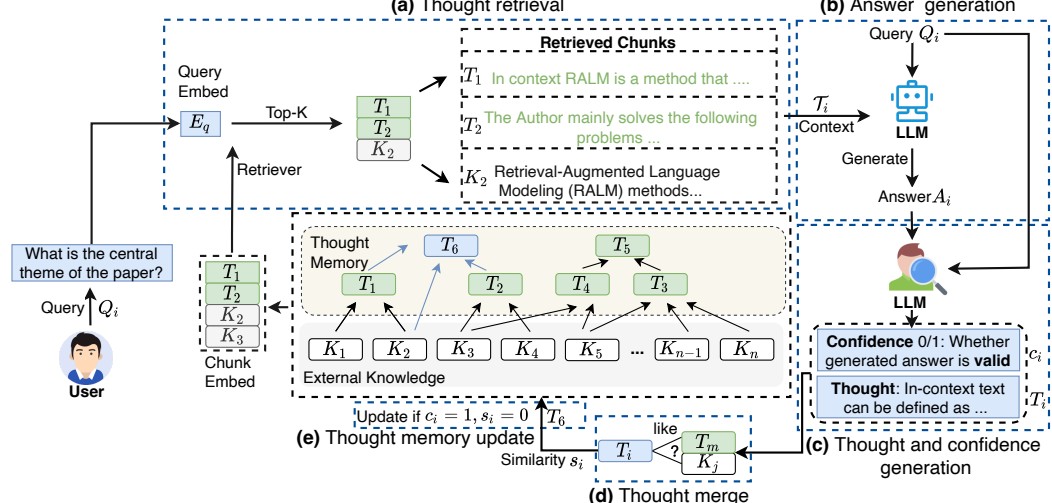

Figure 2: **Thought-Retriever Framework**. **(a) Thought retrieval:** Upon receiving a user query, Thought-Retriever retrieves top-K data chunks from the mixture of external knowledge and thought memory based on embedding similarity; (b) **Answer and confidence generation:** The LLM generates the answer for the user query based on the retrieved data chunks; (c) **Thought generation:** The LLM further generates thought and its confidence based on the user query and the generated answer; (d) **Thought merge:** The calculation of similarity is used to measure whether generated thought will cause redundancy in data chunks; (e) **Thought memory update:** Meaningless and redundant thoughts are removed and the remaining *novel* thoughts are used to update the thought memory.

process, using query $Q_{\text{think}}$, given a set of retrieved data chunks $\mathcal{K}_i$. We define the *source* of an LLM's response, *e.g.*, a thought $T_i$, as the set of data chunks $\mathcal{K}_i$ that are used to generate the response, represented as a mapping $O(T_i) = \mathcal{K}_i$. A key motivation for Thought-Retriever is that an LLM can generate responses based on its past responses; therefore, given a thought $T_i$, we can recursively trace the source of data chunks with mapping $O(\cdot)$, until we find the *root source* via a mapping $\hat{O}(T_i) = \mathcal{K}_i$, consisting of all the raw data chunks from the external knowledge $\mathcal{K}$ that is used to create the thought $T_i$.

## 2.2 MOTIVATING EXAMPLES

To measure how effectively an LLM can utilize external knowledge, we propose to extend the retrieval metric, precision, and recall, with the root source mapping $\hat{O}(\cdot)$. Assuming that answering a user query $Q_{\text{think}}$ requires a set of data chunks $\mathcal{K}_i \in \mathcal{K}$, and an LLM's response is $T_i$. We have

$$\text{Precision} = \frac{|\mathcal{K}_i \cap \hat{O}(T_i)|}{|\hat{O}(T_i)|}, \quad \text{Recall} = \frac{|\mathcal{K}_i \cap \hat{O}(T_i)|}{|\mathcal{K}_i|} \quad (1)$$

As a motivating example, in Figure 1, we assume $\mathcal{K}_i = \{K_1, K_2, K_3, K_4\}$ is required to answer a user query and an LLM can only fit 2 data chunks in its context window. A standard RALM (Figure 1(a)) can achieve perfect precision by retrieving the correct data chunks; however, it has a lower recall since it does not have the context window to hold all the relevant data chunks.

To address the limited context window of RALM, researchers (Chen et al., 2023) proposed hierarchical RALMs (Figure 1(b)), where similar data chunks are summarized into $S_i$ via LLM as a preprocessing step. However, the tree-structured summary structure is rigid, since the summaries $S_i$ are independently generated from the user queries. In Figure 1(b), ideally, chunks $\{K_2, K_3\}$ and $\{K_4, K_5\}$ should be grouped together to answer the user query, where Precision = 1, Recall = 1 could be achieved; however, the tree construction happened before user query, and the generated tree fail to adapt to the diverse future user query.

## 2.3 MOTIVATING EXAMPLES

To stress the above limitations of existing RALMs, as is shown in Figure 1(c), we propose the Thought-Retriever that leverages past LLM thoughts and balances low-level facts and high-level thoughts to answer user queries. In real-world applications, user queries are often sufficiently diverse,

leading to numerous diverse thoughts to meet the demands of new user queries. This valuable observation differentiates Thought-Retriever from existing tree-structured RALMs: (1) Thought-Retriever offers a more flexible structure of thoughts that depends on past user queries, and (2) the thoughts leveraged by Thought-Retriever are byproducts from the standard RALM response, making it easy to implement and brings little computational overhead.

### 2.4 THOUGHT-RETRIEVER FRAMEWORK

**Method Overview.** Figure 2 offers an overview of the proposed Thought-Retriever framework, which consists of four major components: (1) **Thought retrieval**, where data chunks from external knowledge and thought memory are retrieved; (2) **Answer generation**, where an LLM generates the answer for the user query based on the retrieved data chunks; (3) **Thought and confidence generation**, where an LLM further generates thought and its confidence in validation to avoid hallucination based on the user query and the generated answer; (4) **Thought merge**, where similarity is calculated to measure whether generated thought will cause redundancy in data chunks; (5) **Thought memory update**, where meaningless and redundant thoughts are removed; the thought memory is updated with the remaining *novel* thoughts, rather than adopting all the *new* thoughts. We summarize the pipeline of Thought-Retriever in Algorithm 1, whose details are shown as follows. Detailed prompts for this section can be found in Appendix A.2.

---

**Algorithm 1** Thought-Retriever Inference Algorithm

---

**Input:** User queries $\mathcal{Q}$, external knowledge $\mathcal{K}$, thought memory $\mathcal{T}$, language model $L$, retriever $R$ and threshold of similarity $\epsilon$.
**Output:** Answers to user queries $\mathcal{A}$, updated thought memory $\mathcal{T}$.

1: $\mathcal{A} \leftarrow \{\}$
2: **for** $Q_i \in \mathcal{Q}$ **do**
3:     $\mathcal{T}_i \leftarrow R(Q_i, \mathcal{K} \cup \mathcal{T})$ {Thought retrieval}
4:     $A_i \leftarrow L(Q_i, \mathcal{T}_i)$ {Answer generation}
5:     $\mathcal{A} \leftarrow \mathcal{A} \cup A_i$
6:     $T_i, c_i \leftarrow L(Q_i, A_i)$ {Thought and confidence generation}
7:     $s_i \leftarrow \mathbf{1}_{\{\exists j, m \; ; sim(T_i, K_j/T_m) \geq \epsilon)\}}$ {Thought merge}
8:     $\mathcal{T} \leftarrow \mathcal{T} \cup T_i$, if $c_i = 1, s_i = 0$ {Thought memory update}
9: **end for**
10: **return** $\mathcal{A}, \mathcal{T}$

---

**Thought Retrieval.** After receiving a user query $Q_i$, Thought-Retriever $R$ retrieves relevant information $\mathcal{T}_i$ from external knowledge $\mathcal{K}$ and previously generated thought memory $\mathcal{T}$ via embedding similarity ranking. This process is formulated as $\mathcal{T}_i \leftarrow R(Q_i, \mathcal{K} \cup \mathcal{T})$.

**Answer Generation.** Based on the retrieved information $\mathcal{T}_i$, we design a prompt to combine $\mathcal{T}_i$ and user query $Q_i$ and feed the prompt to an LLM $L$ to get the answer $A_i$. It can be articulated as $A_i \leftarrow L(Q_i, \mathcal{T}_i)$.

**Thought and Confidence Generation.** We can generate thoughts via LLM $L$ using the obtained answer $A_i$ and its query $Q_i$ (an example is shown in Table 2.4). However, meaningless thoughts during the generation process may cause hallucinations for LLM and harm performance since some queries may be irrelevant to the external knowledge and thought memory. To solve this issue, we design a special prompt so that LLM $L$ can generate thought $T_i$ and thought quality confidence $c_i$ based on the user's query $Q_i$ and corresponding answer $A_i$. This can be described as $T_i, c_i \leftarrow L(Q_i, A_i)$. Specifically, $c_i$ is a discrete binary value, where 1 indicates that the generated thought $T_i$ is meaningful, and 0 indicates that it is meaningless or hallucinated. This confidence generation is also validated through our experiment in Sec 4.7.

**Thought Merge.** Redundant thoughts may cause LLM to retrieve duplicate information, which is also harmful to the performance of LLM. Therefore, we calculate the similarity $s_i$ between $T_i$ and data chunks $(T_m, K_j)$ to measure whether generated thought $T_i$ will cause redundancy in data chunks. It can be formulated as $s_i \leftarrow \mathbf{1}_{\{\exists j, m \; ; sim(T_i, K_j/T_m) \geq \epsilon)\}}$, where $sim$ denotes embedding similarity based on Contriever (Izacard et al., 2022) (same as retriever used elsewhere in Thought-Retriever) and $\epsilon$ means the threshold of similarity.

Table 1: **Example of thought construction**. For a given query, Thought-Retriever retrieves the corresponding data chunks (thoughts and original data chunks) from the knowledge corpus to get the response. Then Thought-Retriever integrates the query and response into a prompt through the prompt template and obtains the final thought candidate through the output of LLM. This thought candidate is then evaluated to determine whether it is correct and not redundant.

| Query | What has driven significant progress in various NLP tasks in recent years? |
|---|---|
| Response | According to the text, significant progress in various NLP tasks in recent years has been driven in part by benchmarks such as GLUE, whose leaderboards rank models by how well they perform on these diverse tasks. |
| Prompt template | Input: Given query:{query}, given response:{response}. Based on the provided query and its corresponding response, perform the following step: succinctly summarize both the question and answer into a coherent knowledge point, forming a fluent passage. |
| Thought candidate | Here is a summarized knowledge point: In recent years, significant progress has been made in various Natural Language Processing (NLP) tasks. A key driver of this progress is the development of benchmarks, such as GLUE, which provide a standardized way to evaluate and compare the performance of different models on a range of diverse NLP tasks. These benchmarks, which often take the form of leaderboards, rank models based on their performance, fostering competition and innovation in the field. As a result, researchers and developers have been motivated to improve their models, leading to significant advancements in NLP capabilities. |

**Thought Memory Update.** The confidence of thought quality $c_i$ and the similarity $s_i$ determine whether the newly generated thought should be updated into the thought memory $\mathcal{T}$. Here, we design that if the LLM is confident about its answer and the generated thought is not redundant, where $c_i = 1, s_i = 0$, $\mathcal{T}$ will be updated.

# 3 ACADEMICEVAL: NEW BENCHMARK FOR LONG-CONTEXT LLM UNDERSTANDING

Current benchmarks for assessing LLM long-context memory utilization involve tasks such as question-answering, long-context summarization, and classification. Despite being well-constructed, they are limited in flexibility and real-world impact and are costly to acquire. To address these issues, we introduce an innovative benchmark, *AcademicEval*, based on academic papers from arXiv updated daily. *AcademicEval* comes with two datasets: *AcademicEval-abstract* and *AcademicEval-related*.

**AcademicEval-abstract.** This dataset focuses on the summarization of single (*Abstract-single* in Table 2) or multiple (*Abstract-multi* in Table 2) academic papers. The LLM is presented with one or more papers with the abstract and conclusion sections removed and is tasked with writing an abstract. For *Abstract-single*, the generated abstract is directly compared with the paper's original abstract. For *Abstract-multi*, the generated abstract is compared with a summary of abstracts from all the provided papers, which is generated by an expert LLM as a label.

**AcademicEval-related.** This dataset (*Related-multi* in Table 2) introduces a challenging task for assessing an LLM's ability to understand the connections between heterogeneous segments of its long-context memory. The task is to write a related work section based on the title and abstract of a target paper. The LLM needs to use the title and abstract as the query to retrieve memory chunks to complete this task. To be specific, memory chunks depict the abstracts of several papers (each memory chunk corresponds to the abstract of a paper), where some papers are cited in the related work section of the target paper, while others are randomly sampled from the same broader field. The generated related work is then compared to the original related work of the target paper for evaluation.

**Benefits and Contributions.** *AcademicEval* offers several advantages over existing benchmark datasets. *Firstly*, we maintain an up-to-date dataset from arXiv that benefits from the continuous

publication of new papers. This dynamic nature eases overfitting and label leakage problems in static benchmarks and enables the evaluation of LLM self-adaptability. *Secondly*, high-quality labels can be generated with no extra cost as opposed to manually crafted datasets that require human effort. *Thirdly*, our dataset is not only valuable for evaluating LLM but also serves as a practical academic tool in the real world to assist researchers in better understanding their fields and boost productivity. We developed a highly automated codebase for dataset construction that will be released soon. We also launched a public platform that will enable users to easily create similar datasets or utilize LLMs for academic tasks (see details in Appendix F). In addition, the detailed dataset introduction and usage instructions can be found in Appendix A.1 and A.2 respectively.

## 4 EXPERIMENT

### 4.1 EXPERIMENT SETUP

**Additional Datasets.** Besides AcademicEval, we further evaluate Thought-Retriever against state-of-the-art baselines on two public datasets. (1) **GovReport** (Cao and Wang, 2022): This dataset comprises 19,466 reports and associated labels prepared by government research agencies to verify if the LLM is capable of extracting salient words and useful information from a single lengthy governmental document. (2) **WCEP** (Ghalandari et al., 2020): This dataset contains 10,200 entries, each containing multiple news articles associated with an event sourced from the Wikipedia Current Events Portal. It requires the LLM to understand and extract useful information from a cluster of documents.Table 2 summarizes the statistics for all the datasets.

**Baselines.** To gain a comprehensive understanding of our thought retriever's performance on LLM long-term memory tasks, we have adopted several baselines. All experiments with these baselines are conducted under the same LLM: Mistral-8x7B with LLM context length of 4, 096.(Jiang et al., 2024). Note that we set chunk size=500, K=8, $\epsilon = 0.85$, and maximum context length=2,000 tokens for all RALMs. In addition, Contriever is utilized as the retriever in Thought-Retriever, which is verified to be reasonable in Sec 4.5.

First, we consider 2 heuristic-based retrievers: (1) **BM25** (Robertson et al., 2009): A widely-used ranking function in information retrieval. (2) **TF-IDF** (Ramos et al., 2003): A statistical measure that evaluates the importance of a word in a memory. Second, we select 3 deep learning-based retrievers: (3) **Contriever** (Izacard et al., 2022): leveraging contextualized embeddings and neural networks to understand and retrieve relevant memory chunks. (4) **DPR** (Karpukhin et al., 2020): retrieving memory chunks by encoding chunks and queries into dense vectors. (5) **DRAGON** (Lin et al., 2023): employing contrastive learning to train its ability to retrieve memory chunks.

Table 2: **Overview of Datasets:** task types, average length, and number of cases.

| Dataset | Task Type | Avg. len | Cases |
|---|---|---|---|
| **AcademicEval** | | | |
| Abstract-single | Single Sum | 8,295 | 100 |
| Abstract-multi | Multi Sum | 33,637 | 30 |
| Related-multi | Multi Related | 22,107 | 30 |
| **Public Datasets** | | | |
| Gov Report | Single QA | 8,910 | 100 |
| WCEP | Multi QA | 8,176 | 30 |

Third, we consider full context window baselines with document truncation: (6) **Full Context (left)** (Chen et al., 2023): This approach uses the initial segment of a document, truncated to fit within a 4,096-token window. Focusing on the first 4,096 tokens, it prioritizes early content in the document. (7) **Full Context (right)** (Chen et al., 2023): In contrast to Full Context (left), it utilizes the final segment of a document, also truncated to a 4,096-token window. Lastly, we selected two long-context LLMs: (8) **OpenOrca-8k** (Mukherjee et al., 2023) is fine-tuned on the Mistral 7B model using the OpenOrca dataset. At its release time, it was ranked the best model among all models smaller than 30B on Hugging Face, with a maximum context length of 8,192 tokens. (9) **Nous Hermes-32k** (Shen et al., 2023): trained on Mixtral8x7B MoE LLM. It boasts a maximum context length of 32,768 tokens. Note that we do not compare with MEMWALKER (Chen et al., 2023), since it is costly to run and cannot scale to tasks with many data chunks. We use Contriever as Thought-Retriever's retriever.

**Evaluation Metrics.** Our evaluation approach encompasses both traditional metric and AI-based assessments: (1) **F1** (Lin, 2004): This metric computes the semantic similarity between the generated text and the ground truth reference through ROUGE-L (F1). An F1 score closer to 1 indicates a higher alignment with the reference text, signifying the better quality of the generated content. (2) **Win Rate**: Alongside F1, we incorporate feedback from the AI evaluator for a more comprehensive

Table 3: **Thought-Retriever consistently outperforms all the baselines in fact retrieval datasets**. Bold and underline denote the best and second-best results. F1 score evaluates the similarity with the ground truth, higher is better. Win rate compares each method's response with Thought-Retriever, higher is better. Note that the maximum context length is 2,000 tokens for all retriever-based methods and Thought-Retriever employs Contriever as its retriever.

| Type | AcademicEval | | | | | | Public | | | |
|---|---|---|---|---|---|---|---|---|---|---|
| Dataset | Abstract-single | | Abstract-multi | | Related-multi | | Gov Report | | WCEP | |
| Method | F1 | Win Rate | F1 | Win Rate | F1 | Win Rate | F1 | Win Rate | F1 | Win Rate |
| BM25 | 0.212 | 7% | 0.232 | 7% | 0.203 | 40% | 0.211 | 30% | 0.178 | 31% |
| TF-IDF | 0.202 | 4% | 0.225 | 4% | 0.207 | 40% | 0.195 | 35% | 0.223 | 34% |
| Contriever | 0.242 | 13% | 0.232 | 15% | 0.201 | 35% | 0.223 | 40% | 0.211 | 40% |
| DPR | 0.206 | 4% | 0.226 | 4% | 0.196 | 30% | 0.188 | 20% | 0.201 | 33% |
| DRAGON | 0.236 | 7% | 0.226 | 8% | 0.208 | 30% | 0.210 | 40% | 0.231 | 35% |
| Full Context (left) | 0.118 | 2% | 0.155 | 0% | 0.193 | 13% | 0.234 | 45% | 0.207 | 35% |
| Full Context (right) | 0.118 | 1% | 0.149 | 0% | 0.188 | 8% | 0.220 | 40% | 0.210 | 41% |
| OpenOrca-8k | 0.175 | 20% | 0.135 | 3% | 0.135 | 13% | **0.244** | 41% | 0.169 | 30% |
| Nous Hermes-32k | 0.247 | 30% | 0.204 | 7% | 0.183 | 15% | 0.248 | 37% | 0.214 | 37% |
| **Thought-Retriever** | **0.290** | **50%** | **0.275** | **50%** | **0.216** | **50%** | 0.232 | **50%** | **0.238** | **50%** |

assessment. Here, we choose Qwen1.5-72B-chat as our AI evaluator, since it has superb alignment with human preference [1]. This evaluation process involves presenting various responses to the LLM evaluator, who then ranks the quality of the responses. The percentage represents the frequency of a response being chosen over our thought retriever. A rate below 50% suggests that our thought retriever is outperforming the compared baseline.

## 4.2 RETRIEVE CONTEXT FROM FACTUAL KNOWLEDGE

This section is to verify the performance of Thought-Retriever when the external knowledge comes from interaction with facts. We report the performance of our model and baselines in Table 3. Major observations are as follows:

First, in both *AcademicEval* and public benchmarks, Thought-Retriever significantly outperforms most baselines on two metrics. For example, it achieves an average increase of at least 7.6% in F1 score and 16% in win rate across all datasets. This suggests that thoughts formed through interaction with the environment can effectively enhance an LLM's performance in different tasks. Moreover, the comparison and analysis of abstracts generated by different methods on the Abstract-single task (Appendix D) also verify the effectiveness of Thought-Retriever.

Second, we observe that the performance of methods that use the entire text directly have many features on two different benchmarks differs greatly, which contain Full Context baselines and long-context LLMs baselines. However, the performance of retriever-based methods is stable across two benchmarks. This is due to two reasons: (1) AcademicEval is a more challenging benchmark. It contains "multi-modal" information, such as tables, different chapters, different symbol formats, etc. Directly putting this complicated information in

Table 4: **Thought-Retriever can help the LLM quickly learn from other LLMs**. a is the golden setting with real facts and b, c, d are comparative settings without facts.

| Setting | a | b | c | d |
|---|---|---|---|---|
| F1 | 0.25 | 0.19 | 0.22 | 0.24 |

a context makes it difficult for the LLM to process and analyze. For retriever-based methods, they extract the most important information for respond the query from the entire memory, so they can filter out the influence of some redundant information and get better results; (2) Some long-context LLMs may have continuously train on the public benchmarks, which causes the leak of the label and the overfit of the model. In contrast to this, AcademicEval is a good benchmark for evaluating the zero-shot performance of LLM and has no risk of label leakage and overfitting. Since the benchmark is formed using papers from arXiv, it is dynamic and always up-to-date, benefiting from the continuous publication of new papers.

## 4.3 RETRIEVE CONTEXT GENERATED FROM OTHER LLMS

Forming thoughts can be a lengthy process. When a new LLM lacks relevant memory or external knowledge, it is challenging to develop high-quality thought memories from scratch. Consequently,

---
[1]https://qwenlm.github.io/blog/qwen1.5/

we aim to investigate whether Thought-Retriever can help the LLM quickly learn from other LLMs who have already formed expert knowledge.

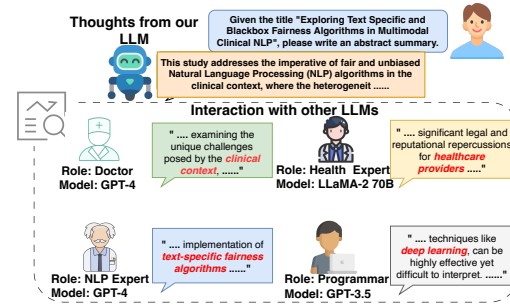

To answer this question, we design an experiment on Abstract-single and the goal of the LLM is to write an abstract summary based on its title. Our LLM builds its memories based on interaction with with other LLMs, which include different roles of an LLM or different LLMs as shown in Fig 3. To verify the effectiveness of Thought-Retriever under this setting, we design four different comparison settings: (a) retrieves knowledge based on the original context of the papers and then uses this knowledge to respond to queries, which serves as a golden setting; (b) feeds the query directly to the LLM to get the responses; (c) let other LLMs provide some relevant data based on a query, then uses this knowledge as raw memories of our LLM, and finally retrieves and gets response based on retriever; (d) utilizes Thought-Retriever to construct thought memories then retrieve thoughts for responding queries based on the setting of (c). We perform an evaluation with metric F1 in 30 cases of Abstract-single, and the results shown in Table

Figure 3: **Thoughts from other LLMs help respond without fact.** It presents an illustrative example in which our LLM communicates with four other LLMs, each an expert in a different field. These expert LLMs are assigned specific roles (e.g., doctor) with different background knowledge. Our LLM is then able to rapidly learn from their thoughts and incorporate them as external knowledge.

4 demonstrate that the rank of them from good to bad is: a, d, c, b. Moreover, the response quality of (d) is very close to that of (a). These observations verify the effect and efficiency of Thought-Retriever when learning from other LLMs. Further results on QA and Reasoning tasks Li et al. (2023) can be found in Appendix H

### 4.4 NEW FINDINGS FROM THOUGHT-RETRIEVER

**Thought Retriever learns to leverage deeper thoughts to answer more abstract user queries.** We conduct a case study to explore the relationship between the abstraction levels of queries and the retrieved information. Specifically, we created a set of questions with varying levels of abstraction and ranked them according to their abstraction level using expert LLM (exact queries can be found in Appendix C). For the retrieved information abstraction level, we first assigned all the raw segments of text from the external knowledge base an abstraction level of 1. The abstraction thought is then calculated as the average abstraction level of all the segments it retrieves, plus one. For example, a thought based solely on the external knowledge base would have an abstraction level of 2. If it also incorporates other thoughts, its abstraction level would be higher. As shown in Fig. 4, where the y-axis represents the abstraction level of the question

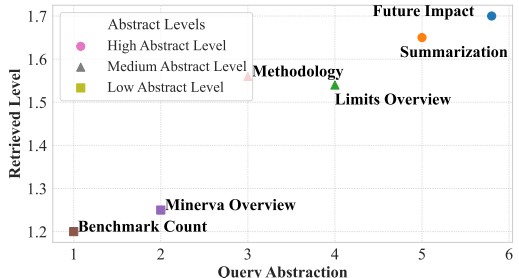

Figure 4: **Deeper thoughts help abstract queries.** This figure illustrates the correlation between six questions, categorized by their level of abstraction as evaluated by expert LLM (x-axis), and the abstraction level of the corresponding retrieved information (y-axis). The questions are grouped into three categories: high abstraction (top 2 questions), medium abstraction, and low abstraction, respectively. Keywords from each question are displayed next to their corresponding data points for clarity.

and the x-axis represents the average abstraction level of all information retrieved by our method. It can be observed that more abstract questions tend to retrieve information with higher abstraction levels.

**Thought-Retriever helps LLM self-evolve after solving more user queries - a new type of scaling law.** To investigate the relationship between the performance of Thought-Retriever and the number of thoughts, we design an experiment using varying numbers of thoughts on Abstract-multi and Related-multi of AcademicEval. As depicted in Fig. 5, there is a distinct trend of increasing F1 scores correlating with the growing number of thoughts, which indicates improved performance.

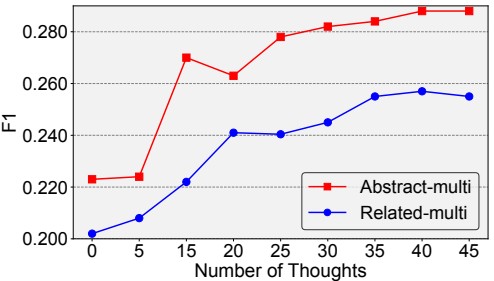
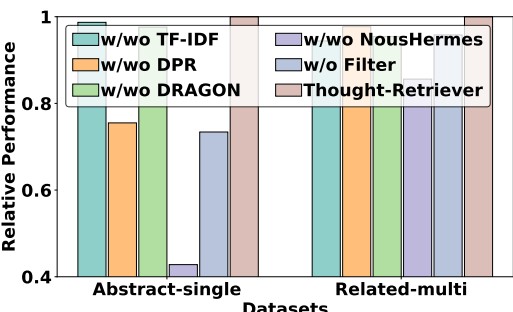

Figure 5: **Thought-Retriever can indeed help LLM self-evolve after solving more user queries.** It illustrates that the performance of LLM across two datasets shows an upward trend as the number of thoughts increases.

Figure 6: **Contriever and thoughts filtering are suitable for Thought-Retriever.** Ablation study of 6 methods on two datasets helps us decide on important components of Thought-Retriever.

Therefore, more interactions with the users enable Thought-Retriever to assist LLMs in self-evolving and developing deeper understandings, demonstrating a new type of scaling law (Kaplan et al., 2020).

### 4.5 ABLATION STUDY

We conduct a series of experiments to investigate the impact of various retrievers. (1) **w/wo TF-IDF/DPR/DRAGON**: In these variants, we replace the retriever (Contriever) in our method with other representative retrievers to assess their effectiveness compared to our current retriever. (2) **w/wo NousHermes**: Here, we substitute the retriever in our method with NousHermes to evaluate its performance relative to our existing retriever-based framework. (3) **w/o Filter**: We remove the confidence generation and thought merge in our framework to assess the importance of filtering meaningless and redundant thoughts.

We report the evaluation results on Abstract-single and Abstract-multi datasets in Fig. 6. **These comparisons clearly show that our method consistently outperforms all the variants**, suggesting that Contriever is most suitable for Thought-Retriever and filtering meaningless and redundant thoughts can bring great improvement to the performance of Thought-Retriever.

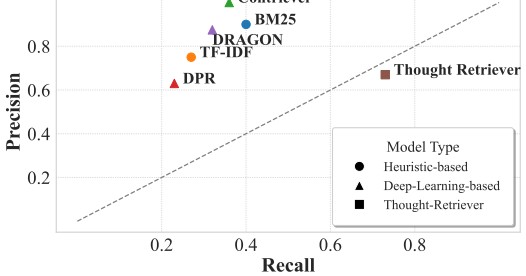

### 4.6 QUALITATIVE ANALYSIS BASED ON PRECISION AND RECALL

In our motivation example in Sec 2.3, we highlighted where traditional methods struggle with recall and precision. Here, using the Related-multi dataset, we show that Thought-Retriever outperforms other baselines in balancing both metrics.

Figure 7: **Thought-Retriever significantly performs better in balancing recall and precision** (The dotted line indicates the exact balance between precision and recall. The closer the dotted line is, the better the balance is). Traditional retriever-based method achieves high precision but low recall. Thought-Retriever balances precision with recall, which maintains good precision when the recall is very high.

In the experiment, the abstracts of the real citations are regarded as ground truth. We aimed to assess how well different retrievers could retrieve information to cover the ground truth, given the limitation of retrieving only 8 chunks of information at a time. We plotted the findings in Fig. 7 where the x-axis is the recall value and the y-axis represents the precision. It can be observed that all traditional retrieval methods displayed significantly low recall values. This is primarily attributed to the top-K retrieval limit since K=8 is far less than the number of ground truth citations. In comparison, Thought-Retriever demonstrates a notable improvement in recall value. This is because it leverages thoughts which is constructed from multiple papers, thereby allowing Thought-Retriever to achieve a much higher recall. More importantly, the Thought-Retriever also exhibits moderately high precision compared to other retrievers. This suggests that, despite a minor trade-off, Thought-Retriever does not significantly compromise its ability to retrieve the most relevant information.

Table 5: **Thought-Retriever demonstrates adaptability across different LLMs**. This table compares Thought-Retriever's performance against baselines on Abstract-single and Abstract-multi tasks using Qwen-7B and Llama-3-70B models. Thought-Retriever consistently delivers the best results, highlighting its adaptability to various LLMs.

| Type | Abstract-single | | | | Abstract-multi | | | |
|---|---|---|---|---|---|---|---|---|
| LLM | Qwen-7b | | Llama-3-70b | | Qwen-7b | | Llama-3-70b | |
| Method | F1 | Win Rate | F1 | Win Rate | F1 | Win Rate | F1 | Win Rate |
| BM25 | 0.196 | 3% | 0.22 | 13% | 0.232 | 7% | 0.233 | 14% |
| TF-IDF | 0.192 | 3% | 0.21 | 17% | 0.220 | 3% | 0.224 | 12% |
| Contriever | 0.231 | 10% | 0.238 | 18% | 0.229 | 4% | 0.228 | 19% |
| DPR | 0.209 | 4% | 0.215 | 18% | 0.222 | 3% | 0.222 | 11% |
| DRAGON | 0.209 | 3% | 0.225 | 13% | 0.224 | 4% | 0.236 | 19% |
| Full Context (left) | 0.069 | 0% | 0.102 | 0% | 0.061 | 0% | 0.107 | 0% |
| Full Context (right) | 0.073 | 0% | 0.104 | 0% | 0.065 | 0% | 0.103 | 0% |
| OpenOrca-8k | 0.175 | 17% | 0.175 | 23% | 0.135 | 17% | 0.135 | 10% |
| Nous Hermes-32k | 0.247 | 20% | 0.247 | 37% | 0.204 | 13% | 0.204 | 7% |
| **Thought-Retriever** | **0.259** | **50%** | **0.285** | **50%** | **0.253** | **50%** | **0.266** | **50%** |

### 4.7 ALGORITHM ADAPTABILITY AND FILTER EFFECTIVENESS

**Thought-Retriever is adaptable to various LLM backbones.** While we use carefully designed prompt templates, Thought-Retriever is not tailored to any specific model. The algorithm is adaptable and effective across various LLM backbones, as shown by the consistent top performance on both Qwen-7B Bai et al. (2023) and Llama-3-70B Dubey et al. (2024) in multiple tasks (Table 4).

**The filter effectively ensures thought quality.** While previous work highlights LLM capabilities such as self-correction Pan et al. (2023), self-refinement Madaan et al. (2024), and self-awareness Pushpanathan et al. (2023), we conducted a human evaluation to verify the reliability of our filter system. Ten volunteers reviewed generated thoughts, with each thought evaluated by five volunteers. The high accuracy—96% on Abstract-single and 93% on Related-multi—demonstrates the effectiveness of the LLM-generated confidence scores.

## 5 ADDITIONAL RELATED WORKS

**(1) Long-context LLMs.** In response to the challenge of long-context processing in LLMs, the most intuitive strategies involve expanding the LLM's context window, enabling it to process longer inputs. These methods include training larger, more advanced models (MosaicML, 2023; LM-SYS, 2023), fine-tuning existing language models to handle wider windows (Tay et al., 2022), and applying positional encoding to extend the context window size (Xiao et al., 2023). However, these methods have shortcomings in their high costs associated with model training and a lack of flexibility, as they do not address the fundamental issue of long context. For instance, to process longer memories, it becomes necessary to engage in additional parameters or model training, which is both rigid and resource-intensive. **(2) Retrieval-Augmented Language Models.** RALM offers a flexible, cost-effective alternative to long-context LLMs by retrieving relevant information from extensive context chunks. Current methods use techniques like token embeddings (Izacard et al., 2022; Lin et al., 2023), keyword searches (Robertson et al., 2009), and fine-tuned rerankers (Ram et al., 2023). Despite promising results, these methods are still limited by LLMs' context capacity, often falling short with growing context lengths. Recently, context summarization methods, such as hierarchical tree structures (Chen et al., 2023), have been proposed to mitigate these limitations. However, these methods are rigid and costly. We address these challenges with a Thought-Retriever framework based on RALM, which summarizes retrieved LLM context segments into thoughts using the user's query.

## 6 CONCLUSION

We propose Thought-Retriever to help LLMs utilize rich external knowledge efficiently, enhancing LLMs by enabling dynamic access without context length limitations. This innovative strategy uses "intermediate thoughts" from past interactions, allowing continuous improvement and understanding. Thought-Retriever demonstrates superior performance across various datasets, including the AcademicEval benchmark, showing its potential to revolutionize AI systems with more adaptive, real-time, context-aware responses. This advancement promises significant improvements in industries like customer service, healthcare, and legal advisory, and lays the groundwork for future research towards achieving more general AI, pushing technology's role in society.

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

## A  DETAILS OF ACADEMICEVAL

In this section, we provide the **data format documentation** for the datasets in our proposed AcademicEval benchmark in Section A.1, and **detailed instructions and prompts** for its usage in Section A.2.

### A.1  DATASET DOCUMENTATION

For AcademicEval-abstract, in the single document setting (Abstract-single), each case includes the paper title, abstract as the label, and main content, excluding the abstract and conclusion. For the multiple document setting (Abstract-multi), each case includes five papers' titles, abstracts, and main contents excluding the abstracts and conclusions. We utilize an expert LLM to summarize the five abstracts of one case into a fluent summary as its label using the prompt in Figure 10. For AcademicEval-related (Related-multi), each paper includes a title, its abstract, its related work as the label, the abstracts of its real citations, and the abstracts of other random papers.

| | Attribute | Description |
|---|---|---|
| Abstract-Single | 'title' | The title of the academic paper. |
| | 'abstract' as label | The abstract of the academic paper. |
| | 'main_content' | The content of the paper excluding the abstract and the conclusion. |
| Abstract-Multi | 'title 1' | The title of the first academic paper. |
| | 'abstract 1' | The abstract of the first academic paper. |
| | 'main_content 1' | The content of the first paper excluding the abstract and the conclusion. |
| | ... | ... |
| | 'title 5' | The title of the fifth academic paper. |
| | 'abstract 5' | The abstract of the fifth academic paper. |
| | 'main_content 5' | The content of the fifth paper excluding the abstract and the conclusion. |
| | 'label' | The summary of five abstracts as a fluent passage. |
| Related-Multi | 'title' | The title of the academic paper. |
| | 'own abstract' | The abstract of the academic paper for wiring related work. |
| | 'own related work as label' | The related work of the academic paper for wiring related work. |
| | 'citations' abstracts' | The abstracts of the target paper's real citations. |
| | 'other random abstracts' | The abstracts of other random papers. |

Table 6: **AcademicEval Dataset Documentation.** This table presents the specific format of the data in our AcademicEval dataset.

### A.2  USAGE INSTRUCTION AND PROMPT UTILIZATION.

Here we offer **detailed instructions** for utilizing the datasets in the AcademicEval benchmark. We also provides **all the necessary prompts we utilized in our experiment.**

**Abstract-Single.**  For the task of single paper abstract summarization, as shown in Figure 8 (a), we first provide a prompt *"Please craft an abstract summarizing the key points from the provided text. The abstract should be of appropriate length and include the main theme, significant findings or arguments, and conclusions of the text. Ensure it captures the essence of the content in a clear, succinct manner"* for the retrieval purpose. We then retrieve information from the paper's main content based on this prompt using a retriever. Then, the LLM would generate an abstract based

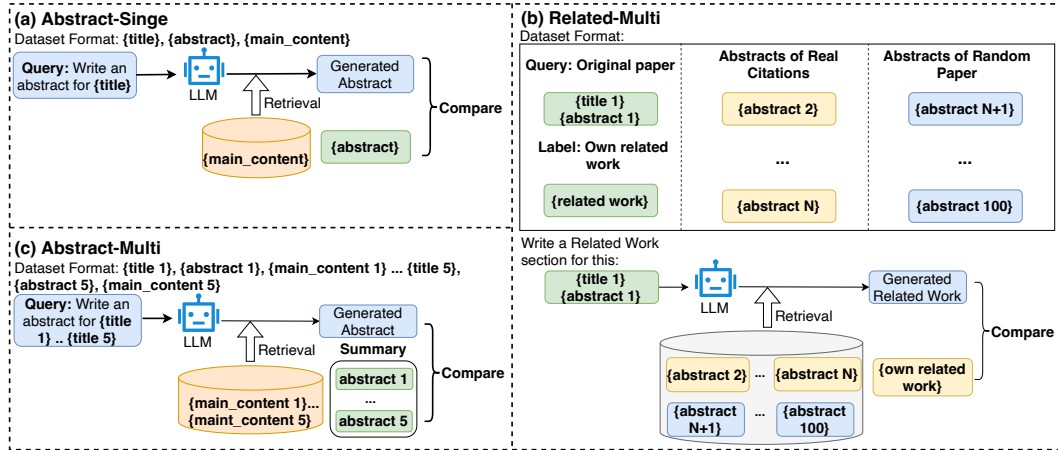

Figure 8: **AcademicEval Usage Instructions.** This figure provides a visualization of the usage instructions for the AcademicEval dataset, as described in Section A.2, to aid understanding.

on the retrieved information using the prompt in Figure 9. Finally, we compare the LLM-generated abstract with the original abstract to do the evaluation.

**Abstract-Multi.** For multiple paper abstracts summarization task, shown in Figure 8 (b), we first provide a prompt *"Please craft an abstract summarizing the key points from the provided text. The abstract should be of appropriate length and include the main theme, significant findings or arguments, and conclusions of the text. Ensure it captures the essence of the content in a clear, succinct manner"* for the retrieval purpose. Then we retrieve information from the main content of the 5 papers based on this prompt. Further, the LLM would generate an abstract based on the retrieved information with the prompt in Figure 9. The generated abstract is compared with the ground truth, which is a summary of the five abstracts created using the prompt in Figure 10.

**Related-Multi.** In the related work task, as shown in Figure 8 (c), we provide the LLM with a prompt *"Could you please write a related work for introducing this paper? Its abstract is: {paper_abs}"*, where *" {paper_abs}"* is sustibute with the paper's real abstract. Following this prompt, the LLM retrieves information from a collection of paper abstracts, comprising the abstracts of real citations in its related work section and random papers. The LLM then generates the related works section based on this retrieved information using the prompt in Figure 11. This generated related work is then compared with the real related work section of the paper to perform evaluation.

# B USER QUERY FORMATION

To model user queries in real-world scenarios for guiding thought generation, we primarily use two approaches: 1) template-based query formation, and 2) LLM-based query formation. The prompts are shown in Figure 12

**Template-based Query Formation.** We construct general and broadly applicable templates for all papers. For example, *"What are the practical applications of the research in 'title'?"* and *"What new perspectives does 'title' offer in its field?"*. During experiments, we substitute *'title'* with the actual paper title to form specific queries.

**LLM-based Query Formation.** Another approach we use to generate more specific queries is by leveraging LLMs. Specifically, we utilize models such as Mistral 8x7B and expert LLM. By providing these models with the paper title and abstract, we ask them to generate diverse questions at varying levels of abstraction. These questions are tailored to each specific paper, allowing for more nuanced and targeted queries.

Please craft an abstract summarizing and connecting the key points from the provided Text.

The text should be composed of content extracted from different papers, potentially spanning varied disciplines, but all addressing overlapping themes or subjects."

The abstract should be of appropriate length (around 300 words), encompassing the main theme, significant findings or arguments, and conclusion of the Text.

Ensure the abstract captures the essence of the content in a clear, succinct manner, providing a coherent summary that bridges the various papers."
Here is the Text: {context}

Figure 9: **Prompt for Writing Abstracts.** This prompt was used in our experiment to ask the LLM to write an abstract based on the retrieved information. We provided in-context instructions to guide the LLM in producing higher-quality responses.

Create a concise, cohesive summary that encapsulates the key points and themes from the following five distinct abstracts. The summary should integrate the main ideas from each abstract to provide a comprehensive overview. It should be about 300 words.

Abstract 1: {abs1}

Abstract 2: {abs2}

Abstract 3: {abs3}

Abstract 4: {abs4}

Abstract 5: {abs5}

Figure 10: **Abstract Multi Ground Truth Prompt.** This prompt was used in our experiment on the Academic-abstract-multi dataset. Specifically, for each data entry, we summarize the abstracts of five papers in the entry to create the ground truth.

## C  SPECIFIC QUERIES OF ABSTRACT LEVEL

This section lists the specific queries utilized in our case study in Section 4.4, demonstrating how Thought-Retriever leverages deeper thoughts for more abstract user queries. Each query is categorized by its general level of abstraction, ranked according to its abstraction level as assessed by an expert LLM and detailed with its exact content in Table 7.

## D  EXAMPLE OUTPUTS COMPARISON OF DIFFERENT METHODS

We present examples of outputs generated using different methods on the AcademicEval-abstract-single dataset. Specifically, in Figure 15, we provide the original paper title and abstract, along with the abstract generated by our Thought-Retriever, **accompanied by a comment from an expert LLM**.

Given the abstract and related work of a research article, along with a sample material, write a paragraph about its related work. Use the following as guidance:

Abstract: This research paper investigates the impact of climate change on global agricultural productivity. The study employs a comprehensive dataset of temperature and precipitation changes over the past century, combined with historical crop yield data. Through advanced statistical modeling and machine learning techniques, the research identifies significant correlations between temperature and precipitation fluctuations and variations in crop yields. Furthermore, it predicts future scenarios of agricultural productivity under different climate change scenarios, providing valuable insights for policymakers and stakeholders in the agricultural sector to develop adaptive strategies.

Related Work: Previous studies in the field have explored the relationship between climate change and agriculture but have primarily focused on specific regions or crops. Smith et al. (2017) conducted a comprehensive analysis of the impact of temperature on wheat yields in North America, highlighting the vulnerability of wheat crops to warming temperatures. Additionally, Johnson et al. (2019) investigated the effects of changing precipitation patterns on rice production in Southeast Asia, emphasizing the importance of water management in mitigating climate-related risks to agriculture. While these studies contribute valuable insights, our research extends their scope by considering a global perspective and employing advanced modeling techniques to provide more accurate predictions of future agricultural productivity under climate change scenarios.

Based on the abstract of this article and related materials, write a paragraph about its related work:
Abstract: {abstract}
Related materials: {context}

Figure 11: **Prompt for Writing Related Works.** This prompt was used in our experiment to ask the LLM to write a related work section based on the original paper's abstract and the retrieved related materials. We also provided an example of in-context learning to enable the LLM to perform more effectively on this challenging task.

In Figure 16, we show abstracts generated using DPR and TF-IDF, also **accompanied by expert LLM comments** for comparison. In Figure 17, we showed the example abstract generated by the long context model Nous Hermes 32k and the **corresponding comments from the expert LLM.** It is evident that the abstract generated by our Thought-Retriever is more comprehensive and coherent, with better management of specification and abstraction levels. Below, we include a comprehensive comment from the expert LLM:

"***The Thought-Retriever abstract is the best and most aligned with the original abstract.*** *It effectively* ***captures all key points***, *including the critique of leaderboard metrics and the need to consider factors beyond accuracy, such as energy efficiency, model size, and inference latency. It also calls for increased transparency on leaderboards, emphasizing a holistic approach to NLP evaluation that includes practical statistics to provide a comprehensive measure of model utility. This abstract is* ***clear, well-organized, and includes a call to action*** *for changes in leaderboard reporting to better serve the practical needs of NLP practitioners.*"

---

Template-based Query Formation:
What new perspectives does '{title}' offer in its field
How might the findings in '{title}' influence future research?
What are the practical applications of the research in '{title}'?
In what ways does '{title}' challenge existing theories or beliefs?
How does '{title}' contribute to our understanding of its subject matter?
What does the statement '{sentence}' imply in the context of '{title}'?
How does the sentence '{sentence}' relate to the overall theme of '{title}'?

LLM-based Query Formation:
Given the paper title: {title}; and its abstract {abstract}, please ask 20
questions that would be helpful for writing its related work section. Each
questions should have a number at the begining. For example:\n 1.<Put
Your Question Here>\n2. <Put Your Question Here>, etc. The questions
should be diverse and with different level of abstraction.

---

Figure 12: **User Query Formation Prompt.** This figure presents the prompt used to model real-world user queries. Specifically, it includes two methods: template-based query formation, where general question templates are created to be suitable for a wide range of papers, and LLM-based query formation, where this prompt is used to ask an LLM to generate diverse queries.

---

Input: Given question:{question}, given answer:{context}. Based on the
provided question and its corresponding answer, perform the following
steps:

 Step 1: Determine if the answer is an actual answer or if it merely indicates
that the question cannot be answered due to insufficient information. If the
latter is true, just output '0' without any extra words, otherwise output '1'.

 Step 2: If it is a valid answer, succinctly summarize both the question and
answer into a coherent knowledge point, forming a fluent passage.

---

Figure 13: **Thought and Confidence Generation Prompt.** This prompt is used for Thought and Confidence Generation as described in Section 2.4. It evaluates whether the answer is valid and meaningful, and then summarizes the query and answer into a thought.

"*In contrast, the **DPR and long context model abstract, while touching on similar points, is less comprehensive and focuses more on specific suggestions** like user-specific leaderboards and revealed preference theory without fully encapsulating the broader argument about the divergence between leaderboard metrics and practitioner needs. The **TFIDF abstract diverges the most**, discussing related topics like brittleness, bias, and out-of-distribution data, but it **does not focus specifically on the central argument** about leaderboard metrics versus practical utility, making it less aligned with the original abstract's intent.*"

# E    DISCUSSION

**Transformative Impact and Real-World Applications.**    The Thought-Retriever represents a paradigm shift in AI systems, transforming them from static repositories of knowledge to dynamic, intelligent frameworks that interact and learn. Its unique architecture not only processes and retrieves information but also evolves with each user interaction, effectively 'thinking' and adapting over time. Such an intelligent system is crucial for scenarios where real-time learning and context-aware

Given the original abstract:{original},and given the two generated abstracts:

Generated Abstract 1:{gen1}; and Generated Abstract 2: {gen2}, plase evaluate which one is closer to the original abstract.

Just output 'Abstract 1 is better' or 'Abstract 2 is better', no extra words.

Figure 14: **AI Evaluation Prompt.** This prompt is used for the AI Evaluation metric Win Rate, as described in Section 4. Given two generated answers and the ground truth answer, we ask the expert LLM to determine which generated answer aligns more closely with the ground truth.

| Abstraction | Rank (expert LLM) | Query |
| --- | --- | --- |
| High | 6 (Most Abstract) | "What are the broader future implications of user-centric utility in NLP model evaluation?" |
| High | 5 | "Please craft an abstract summarizing the key points from the provided text." |
| Medium | 4 | "What are some of the limitations of this study?" |
| Medium | 3 | "What are the key methods introduced in this paper?" |
| Low | 2 | "Please explain the term Minerva to me." |
| Low | 1 (Least Abstract) | "How many benchmarks are used to test the model's long context understanding ability in this paper?" |

Table 7: **Sample Queries Used in Abstraction Level Case Study.** This table presents sample queries from the case study conducted in Section 4, which demonstrates how Thought-Retriever learns to leverage deeper thoughts to answer more abstract user queries.

responses are vital. For instance, existing AI service systems could be significantly enhanced by incorporating our approach. By storing original guidelines and regulations as part of the external knowledge base and recording each human query and its results as thoughts, these systems can evolve into more intelligent entities capable of continuous improvement and learning. This adaptive capability makes the Thought-Retriever an invaluable tool for dynamic and ever-changing industrial environments, where quick decision-making based on historical data and evolving information is crucial. In sectors like customer service, healthcare, and legal advisory, where personalized and informed responses are key, the Thought-Retriever can provide more accurate, context-aware, and efficient solutions. Its ability to continuously learn and adapt from user interactions positions it as a groundbreaking tool for transforming how industries interact with and utilize AI technology.

**Future Research.** Inspired by human thinking, our Thought-Retriever represents a solid step toward general AI agents. Building on this foundation, future research could address several key challenges. Firstly, scalability and efficiency in processing increasingly complex datasets will be crucial. This involves not only enhancing computational power but also refining algorithms for greater precision and speed. Secondly, understanding and mimicking human-like reasoning remains a pivotal goal. This includes grasping nuances in language, emotion, and cultural contexts, and pushing the boundaries of what AI can comprehend and respond to. Moreover, ensuring ethical considerations in AI decision-making is significant. As the retriever evolves, its impact on privacy, security, and societal norms must be rigorously evaluated and guided. Finally, explore new domains of application, such as personalized education, mental health analysis, and advanced robotics.

**Original Abstract:**

Benchmarks such as GLUE have helped drive advances in NLP by incentivizing the creation of more accurate models. While this leaderboard paradigm has been remarkably successful, a historical focus on performance-based evaluation has been at the expense of other qualities that the NLP community values in models, such as compactness, fairness, and energy efficiency. In this opinion paper, we study the divergence between what is incentivized by leaderboards and what is useful in practice through the lens of microeconomic theory. We frame both the leaderboard and NLP practitioners as consumers and the benefit they get from a model as its utility to them. With this framing, we formalize how leaderboards – in their current form – can be poor proxies for the NLP community at large. For example, a highly inefficient model would provide less utility to practitioners but not to a leaderboard, since it is a cost that only the former must bear. To allow practitioners to better estimate a model's utility to them, we advocate for more transparency on leaderboards, such as the reporting of statistics that are of practical concern (e.g., model size, energy efficiency, and inference latency).

**Abstract by Thought Retriever:**

The paper "Utility is in the Eye of the User: A Critique of NLP Leaderboards" critiques the prevalent assumption that leaderboard metrics accurately reflect the utility of NLP models. It emphasizes the need to consider multiple factors beyond predictive accuracy, including energy efficiency, model size, and inference latency, when evaluating NLP models. The paper argues that leaderboard values often diverge from the values of practitioners, calling for increased transparency on leaderboards to better align these metrics with the practical concerns of the NLP community. It highlights the importance of a holistic approach to NLP evaluation, advocating for the inclusion of practical statistics on leaderboards to provide a more comprehensive measure of model utility. The research suggests that this shift towards transparency and comprehensive evaluation will encourage the development of models that are not only accurate but also efficient and fair, ultimately better serving the needs of diverse users in the NLP community.

. . . . . . . . . . . . . . . . . . . . . . . . . . . . . . . . . . . . . . . . . . . . . . . . . . . . . . . . . . . . . . . . . . . .

**Comments from Expert LLMs:**

This abstract effectively **captures all critical aspects** of the original while emphasizing the divergence between leaderboard metrics and practical utility. It **maintains clarity and completeness**, making it an excellent summary of the paper's main arguments. Specifically, it includes key elements such as energy efficiency, model size, and the need for a holistic approach, which are well-aligned with the original abstract. The Thought Retriever abstract is comprehensive and aligns well with the original content, making it **the best among the generated abstracts.**

Figure 15: **Qualitative Example - Original Abstract and Abstract Generated by Thought-Retriever.** This figure presents example outputs from different methods using data from the AcademicEval-abstract-single dataset. Specifically, it shows the original abstract alongside the abstract generated by Thought-Retriever, accompanied by a comment from an expert LLM. Comparison examples generated by DPR and TF-IDF are shown in Figure 16, while comparison examples by the long context model can be found in Figure 17.

# F  ARXIV COPILOT DEMO

Based on the Thought-Retriever, we further propose a demo named Arxiv Copilot and deploy it on the huggingface shown in Figure 18, which aims to provide personalized academic service. More

Table 8: **Thought Retriever's Effectiveness on QA and Reasoning Tasks.** This table presents results from LooGLE, a recent and widely used QA and reasoning benchmark. Our Thought Retriever consistently outperforms all other baselines.

| | BM25 | TF-IDF | Contriever | DPR | Dragon | Full Context (left) | Full Context (right) | OpenOrca - 8K | Nous Hermes-32K | Thought Retriever |
|---|---|---|---|---|---|---|---|---|---|---|
| QA Accuracy | 10% | 13% | 20% | 17% | 13% | 7% | 7% | 0% | 3% | **27%** |
| Reasoning Accuracy | 30% | 33% | 50% | 20% | 27% | 20% | 13% | 10% | 17% | **57%** |

specifically, it consists of three main parts as below. Firstly, in the first "Profile" part, users can enter the researcher's name and generate a research profile. Secondly, in the research trend part, users can select a time range and get relevant topic trends and ideas. Finally, in the "Chat and Feedback" part, users can Chat with Arxiv Copilot and choose the better response from two answers. Here we appreciate any further feedback.

**Profile**   In this part, as shown in Figure 18 (a), user can input his/her name in a standard format to get the profile from arxiv here.

**Research Trend**   As shown in Figure 18 (b), Arxiv Copilot will give the user personalized research trends and ideas if the user has set his/her profile. Otherwise, general research trends will be provided.

**Chat and Feedback**   As shown in Figure 18 (c), each time Arxiv Copilot will give two answers. If the user prefers the second answer, he/she can click 'like' below the second answer and the first answer will be removed. If the user clicks 'dislike', the second answer will be removed.

## G   LIMITATIONS

Despite the promising results and contributions of our work, we would like to discuss some limitations. Our experiments and the AcademicEval dataset primarily utilize papers from AI-related fields, which could limit the generalizability of our findings. Future work should consider extending the scope to a broader range of disciplines.

Additionally, our experiments and evaluations are conducted in English. This focus on English may overlook the nuances and challenges associated with other languages. Expanding our approach to include multilingual datasets and evaluations could provide a more comprehensive assessment of its effectiveness.

While AcademicEval provides a dynamic and continuously updated dataset from arXiv, it is reliant on the availability and quality of the papers uploaded to the platform. We assume and hope that researchers will continue to produce novel and high-quality work.

Lastly, while our framework shows effectiveness in our experiments, its robustness, scalability, and adaptability to real-world, extremely large-scale applications have yet to be fully tested. We are actively working on our demos and hope to provide more exciting updates on this front in the near future.

## H   FURTHER RESULT ON QA AND REASONING TASK

We evaluated Thought-Retriever on the recent LooGLE dataset Li et al. (2023) for QA and reasoning tasks. As shown in the Table 8, it consistently outperformed all baselines, demonstrating strong performance in both QA and reasoning accuracy.

## I   API ACKNOWLEDGEMENT

We used Together AI's API to conduct our experiments. There are no specific requirements to run our code. Essentially, our experimental setup can be replicated by anyone with standard laptops or desktop computers and any compatible API, not necessarily Together AI's API.

**Abstract by DPR:**

The text discusses the importance of including costs like model size, energy efficiency, training time, and inference latency in leaderboard rankings for NLP models, which currently prioritize accuracy above all else. This focus on accuracy can result in models that provide high utility to leaderboards but not to practitioners, as they may come with unaccounted costs. The text suggests that model creators could voluntarily submit these cost statistics, but without a requirement, there is no explicit incentive to do so. It proposes creating user-specific leaderboards with desired thresholds and reporting performance on the worst-off group to incentivize improvements. The use of revealed preference theory is suggested to estimate a model's utility by examining its real-world use. The text also advocates for making usage statistics public and allowing users to adjust metric weightings to create personalized leaderboards, aligning rankings with practical utility.

**Comments from Expert LLMs:**

This abstract is detailed and covers the essential points but tends to be **repetitive and slightly verbose.** It effectively communicates the need for including various costs in leaderboard rankings, but it **lacks the concise clarity** found in the Thought Retriever abstract. Additionally, the abstract is somewhat **incoherent** due to the repetitive structure and lack of clear flow between ideas. It does **not explicitly mention the holistic approach to evaluation or the practical applications of this research**, which are important elements from the original abstract.

**Abstract by TFIDF:**

Recent research has focused on the evaluation and development of NLP models, highlighting issues such as brittleness, bias, and lack of robustness. Datasets like SQuAD 2.0 have addressed challenges like unanswerable questions, while biases in NLP systems lead to better performance on gender-stereotypical inputs and inanimate objects. A key issue in evaluation is aligning priorities between leaderboard rankings and practitioners' preferences. While accuracy is valued by both, practitioners also consider out-of-distribution data, fairness, and real-world applicability, which leaderboards often ignore. The study suggests that leaderboard design should include prediction costs and fairness measures to better align with real-world requirements. Alternative evaluation methods that reflect the complexities of NLP model usage are also recommended.

**Comments from Expert LLMs:**

This abstract is concise and well-organized, addressing the key points effectively. It communicates the need for better alignment between leaderboard rankings and practical utility but **does not delve as deeply** into specific suggestions for improvement as the Thought Retriever abstract. While it provides a detailed discussion on issues like brittleness and bias, it might be **too detailed**, potentially overwhelming the reader with **too many aspects without a clear focus on the main argument of the original abstract.** Additionally, it **lacks specific elements** like the advocacy for increased transparency and the detailed explanation of how leaderboards can be poor proxies for the NLP community, which are crucial parts of the original abstract.

Figure 16: **Qualitative Example - Abstracts Generated by DPR and TF-IDF.** This figure presents example outputs using data from the AcademicEval-abstract-single dataset, generated by traditional methods: DPR and TF-IDF. We also include comments from an expert LLM. The original abstract and the abstract generated by our Thought-Retriever can be found in Figure 15.

**Abstract by Nours Hermes - 32k:**

The text addresses the limitations of performance-based evaluation in NLP tasks and advocates for more comprehensive leaderboard designs. It points out the disparity between leaderboard priorities and the values of the NLP community, such as fairness, energy efficiency, and robustness. By viewing leaderboards and NLP practitioners as consumers of models, the text suggests using utility functions to align their preferences. Three main limitations are identified: non-smooth utility, prediction cost, and robustness. Leaderboards only increase utility from accuracy improvements that affect rankings, while practitioners value consistent accuracy gains. Leaderboards ignore prediction costs, such as model size, energy efficiency, and inference latency. They also overlook robustness, crucial for real-world applications. To address these issues, the text proposes that leaderboards require transparency, reporting statistics like training time, model size, inference latency, and energy efficiency. This transparency would help practitioners estimate model utility and re-rank models based on their needs. The text also suggests personalized leaderboards, allowing users to adjust metric weightings or creating leaderboards tailored to different user types based on their preferences or actual usage.

........................................................................................................

**Comments from Expert LLMs:**

The long context model's abstract is detailed but **overly verbose**, making it **harder to quickly grasp the core arguments**. In contrast, the thought retriever's abstract is more succinct and focused, effectively summarizing the critique of leaderboard metrics and the need for comprehensive evaluation factors. This makes the thought retriever's abstract clearer and better aligned with the original abstract's intent.

Figure 17: **Qualitative Example - Abstracts Generated by Long Context Model.** This figure presents example outputs using data from the AcademicEval-abstract-single dataset, generated by the long context model Nours Hermes - 32k. The original abstract and the abstract generated by our Thought-Retriever can be found in Figure 15.

(a) Profile.

(b) Research Trend.

(c) Chat and Feedback.

Figure 18: **Arxiv Copilot Demo.** This figure shows the demo built based on our proposed Thought-Retriever, which is publicly available on Hugging Face. It offers personalized academic services, aiming to test the real-world robustness of our algorithm and provide social benefits.

