# OpenReview forum: "Thought-Retriever: Don’t Just Retrieve Raw Data, Retrieve Thoughts"
_ICLR.cc/2025/Conference — ICLR 2025 Conference Withdrawn Submission_

### Official Review · Reviewer_chJn · 2024-10-20

**Soundness:** 2
**Presentation:** 2
**Contribution:** 2
**Rating:** 3
**Confidence:** 4

**Summary:**

This study proposed ThoughtRetriever, a novel model-agnostic algorithm that helps LLMs generate output conditioned on arbitrarily long external data. The method enables the self-evolution of LLMs during continuous interactions. Experimental results show that the method could enhance the model performance compared with traditional RAG system, together with a new benchmark AcademicEval for ultra-long context.

**Strengths:**

1. The method is intriguing and provides a self-evolving solution to enhance Large Language Models (LLMs) through thought generation, merging, and updating.

2. Extensive experiments demonstrate the effectiveness of the proposed method, which achieves state-of-the-art performance compared to baseline models.

3. ThoughtRetriever outperforms other baselines in balancing both precision and recall, whereas other baselines fall short in recall.

4. The proposed benchmark, based on the up-to-date arXiv dataset, is useful and holds promise for facilitating future research.

**Weaknesses:**

1. The study proposes the question how to effectively and efficiently help LLMs utilize (arbitrarily) rich external knowledge, and then propose ThoughtRetriever. But how do your prove the significant efficiency of your work? It should be proved through experiments since there is also extra computation cost of your method such as response generation, merging and uploading.  Furthermore, I believe the study could propose a more intuitive question of previous studies such as low recall because of some essential problems in previous studies.

2. The proposed metrics, Precision and Recall in Eq (1), are confusing. Why use the response $T_i$ for calculation? I suppose that if $d_1$ and $T_1$ are retrieved (here, I use $d_1$ to represent a single document chunk and $T_1$ for the retrieved thought from the memory), we should calculate the intersection of $d_1$, $T_1$ with $K_i$. It appears that the calculation only considers $T_1$ to $K_i$. The terminology is significantly confusing; as mentioned, you use $K_i$ to denote both a single data chunk and a set of data chunks, and $T_i$ for both thought and response.

3. If the assumption in W1 is correct, and you use $T_1$ to represent a set of documents, such as $O(T_1) = {K_2, K_3, K_4}$, then the recall metric could be enhanced since the numerator of recall becomes larger due to different granularity levels. As described in Section 4.4, the recall is significantly higher than with other methods, making the comparison seem unfair. In other words, how can you ensure that $T_1$ accurately represents the knowledge in $K_2, K_3$, and $K_4$? A detailed analysis would be beneficial.

4. Importanct baselines are missing. The proposed method is more closely related to Hierarchical RALMs, as mentioned in the related work. However, the experiments only compared it with some sparse and dense retrievers. Additional comparisons with works such as [1-3] should be included to prove the effectiveness of this method. These work considers the summarization of the retrieved documents for downstream tasks.

5. Experiments involving state-of-the-art models such as GPT-4 or Claude-3 are lacking, which could further demonstrate the effectiveness of the ThoughtRetriever.

6. The hyper-parameter $\epsilon$ seems to be very sensitive. How do you determine such a parameter, and how do you evaluate its impact? I suggest conducting an experiment to demonstrate the relationship between similarity and generated content.

7. A human experiment to evaluate the effectiveness of the metric WinRate in your task could be beneficial.

8. Moreover, how do you ensure the quality of your proposed benchmark, which are generated using LLMs?


[1] Walking down the memory maze: Beyond context limit through interactive reading.
[2] RAPTOR: Recursive Abstractive Processing for Tree-Organized Retrieval.
[3] GraphRAG: Unlocking LLM discovery on narrative private data.

**Questions:**

Refer to Weakness.
I would be happy to revise my evaluation if these concerns can be addressed.

---

### Official Review · Reviewer_ehHC · 2024-11-01

**Soundness:** 1
**Presentation:** 2
**Contribution:** 2
**Rating:** 3
**Confidence:** 4

**Summary:**

The authors propose to retrieve the thoughts (generated from the query and predicted answer), instead of retrieving only the raw textual chunks from external documents, and then to use those retrieved thoughts as well as raw textual chunks to generate the answer. Also, to demonstrate the efficacy of the proposed method, the authors introduce two benchmark datasets, whose goal is to generate the abstract or the related work given the paper from arXiv. Then, the authors show that the proposed approach outperforms other baselines on it along with other existing benchmark datasets.

**Strengths:**

* The idea of generating the answer based on the thoughts retrieved from the thought pool is interesting.
* The approach to collect the thoughts and to maintain the quality of the thought pool is convincing.
* The proposed method outperforms relevant baselines.

**Weaknesses:**

* The design of the proposed benchmark dataset is not clear. Specifically, what is the input of the model to generate the abstract or the related work (is this the whole paper)? What if the number of tokens in the related work section is larger than the number of tokens that LMs can generate? In addition to them, the authors use the LMs with a maximum token length of less than 10K, and it is unclear how to provide the LMs with the whole (very long) paper to generate the target output (abstract or related work).
* In addition to this, the quality of the proposed benchmark dataset is questionable. Specifically, the target output for it is either abstract or related work of the paper, and (even if there is one claim that the paper wants to make) there are multiple ways to write abstracts and related works, which further depends on the style of authors. To confirm its quality as the benchmark dataset, more analyses are required: for example, the authors can ask another human annotator to write an abstract (or a related work section) given the input that the model is provided with, and compare it with the ground-truth one. If there are high-agreements between two different abstracts, we can confirm that there is less variety in writing an abstract, from which we can use the abstract written in the paper as the ground-truth label. Otherwise, it is questionable whether we can believe that the abstract written in the paper can be considered as one single ground-truth label.
* There is a previous work [1] that utilizes the rationale (a kind of thoughts) in retrieval, instead of using the raw textual chunks (which is similar to the proposal of this work). It may be worthwhile to discuss.
* It is unclear why the win rate of the proposed model across all datasets is set to 50%. Could you clarify more on how to calculate the win rate?

---

[1] Knowledge-Augmented Reasoning Distillation for Small Language Models in Knowledge-Intensive Tasks, NeurIPS 2023.

**Questions:**

Please see the weaknesses above.

---

### Official Review · Reviewer_XJUU · 2024-11-03

**Soundness:** 3
**Presentation:** 3
**Contribution:** 3
**Rating:** 6
**Confidence:** 3

**Summary:**

Thought-Retriever is a model-agnostic retrieval algorithm designed to assist Large Language Models (LLMs) in overcoming context length limitations and effectively integrating and utilizing vast external knowledge. This algorithm leverages the Thoughts generated from LLMs' past responses, filters and organizes them, and then combines them with external knowledge to process new user queries and generate more comprehensive results. Beyond enhancing performance of LLMs, it can also self-evolve by continuously enriching its Thought memory as it processes more user queries. Additionally, the paper introduces a new benchmark, AcademicEval, which evaluates LLMs' ability to understand and summarize using ultra-long contexts. Experimental results show that Thought-Retriever not only performs well across three datasets but also demonstrates potential in handling more abstract queries and self-evolution. Ablation studies and qualitative analyses further confirm the importance of key components in the Thought-Retriever design.

**Strengths:**

- **Originality**: This paper proposes an innovative retrieval framework that leverages past responses generated by LLMs to transform them into storable and retrievable "Thoughts" to assist with new queries. It also develops a dynamic benchmark, AcademicEval, which challenges the ability of LLMs to handle ultra-long contexts and mitigates issues of overfitting and label leakage in static benchmarks.
- **Quality**: The experimental design in the paper is rigorous, employing multiple datasets and baseline methods for comparison. The experimental results demonstrate that Thought-Retriever generally outperforms the existing nine baseline methods in retrieval-based question answering.
- **Clarity**: The paper is well-structured, with a clear logical flow from problem statement to methodology, experiments, and conclusion. Each component of the Thought-Retriever framework is detailed, including algorithmic pseudocode and framework diagrams.

**Weaknesses:**

- Although Thought-Retriever is conceptually innovative, it relies heavily on the performance of existing retrieval algorithms and LLMs.
- When generating Confidence for Thoughts, a Prompt is used for binary classification, which results in a non-continuous Confidence score and may introduce inaccuracies in the judgment.
- There is an error in the average increase mentioned on line 352; it should not be 7.6%.
- The text on line 506 should correspond to Table 5, but the reference in the text mistakenly points to Table 4.

**Questions:**

- How is the average increase calculated on line 352, and could there be a calculation error?
- The Thought-Retriever updates continuously during program execution. What kind of data structure is it stored in? Are Thoughts and other text chunks stored in memory, or have they been locally stored?
- How does the retrieval process work for this tree-like structure? Or is the tree-like relationship implicit, and retrieval performed sequentially?

---

### Official Review · Reviewer_xuh4 · 2024-11-04

**Soundness:** 2
**Presentation:** 2
**Contribution:** 2
**Rating:** 3
**Confidence:** 5

**Summary:**

The paper introduces Thought-Retriever, focusing on addressing limitations in current large language model approaches. The authors identify two key motivational challenges: First, while long-context LLMs attempt to expand context windows, they cannot fundamentally address interactions with ultra-rich external knowledge due to quadratic computational complexity relative to context length. Second, while retrieval-augmented LLMs can access external knowledge bases through retrievers, they remain constrained by context length limits, only able to retrieve top-K raw data chunks that fit within these bounds.
The methodology centers on Thought-Retriever, an LLM-agnostic self-evolving retrieval framework that leverages historical LLM responses to handle new queries. The framework's key insight is enabling LLMs to utilize their intermediate responses from past queries, which they term "thoughts." These thoughts undergo filtering to remove meaningless and redundant content before being organized into a hierarchical memory structure.
For evaluation, the authors use two public datasets: GovReport and WCEP. Moreover, they introduce AcademicEval, a novel benchmark that tests LLMs' ability to utilize long context in solving tasks about arXiv academic papers. AcademicEval includes two datasets: one focused on abstract generation (both single-document and multi-document summarization), and another for related work section generation.
The experimental results demonstrate that Thought-Retriever consistently outperforms state-of-the-art retrieval-augmented and long-context baselines. Specifically, across various tasks, it achieves an average increase of at least 7.6% in F1 score and 16% in win rate. Further experiments show that Thought-Retriever enables LLM self-evolution through increased query solving experience, and it learns to leverage deeper thoughts for more abstract queries.

**Strengths:**

- The paper presents well-defined motivations addressing the limitations of current LLMs, particularly their context window constraints and inefficient use of external knowledge. The goals aim for meaningful advances in LLM capabilities through self-evolution, representing an ambitious yet practical direction for the field.
- Introducing thought memorization and re-exploitation as a self-evolving mechanism represents an interesting conceptual advance. This approach suggests a new type of scaling law where model capabilities grow through accumulated experience rather than just parameter count or context length, offering a more sustainable path to improvement.
- The combined generation of thought and confidence scores is based on the user query and the generated answer.
AcademicEval is centred on the continuous publication of new papers (dynamic nature) for self-adaptability evaluation. The evaluation of the related work generation task is an interesting testbed for understanding the LLM's ability to reuse its previously generated thoughts.
- The evaluation includes comparisons with models having different context windows.
- Incorporating LLM-as-a-judge evaluation, including metrics like Win Rate and query abstraction assessment, provides a more nuanced understanding of model performance beyond traditional metrics.
- The investigation of the relationship between query abstraction and the hierarchical position of retrieved thoughts (Section 4.4) offers valuable insights into how the system organizes and utilizes knowledge at different levels of abstraction.
- Inclusion of ablation studies about the contribution of each main component to the overall framework performance.

**Weaknesses:**

- When examined closely, the core contribution primarily consists of enriching a tree-structured external memory with silver "rationales" during question answering through prompt-based approaches only. This represents a modest technical advancement, essentially implementing a structured memory system with LLM-generated content.
- The presentation quality shows noticeable deficiencies, with multiple typographical errors throughout the manuscript (see the section below). The writing quality suggests insufficient attention to detail in the manuscript preparation. This level of presentation does not meet the standards expected for a top-tier venue like ICLR, as it impacts the clarity of the technical contributions.
- The paper contains unsupported claims that require empirical evidence. For example, "In real-world applications, user queries are often sufficiently diverse, leading to numerous diverse thoughts to meet the demands of new user queries" is presented without supporting data or citations. Such severe claims require concrete evidence through user studies, empirical analysis, or references to existing literature.
- The implementation choices for the retriever component require better justification. The use of Contriever (L215) appears suboptimal given current MTEB rankings. While the authors reference experimental findings in L296, the selection of baselines does not align with current state-of-the-art approaches. This choice needs stronger justification or consideration of more recent alternatives to strengthen the comparative analysis.
- The use of LLMs for generating multi-document summarization targets (L258-259) raises methodological concerns. This approach requires careful consideration regarding dataset quality, trustworthiness and community impact.
- The dataset documentation presents significant gaps that affect the paper's scientific rigour. The authors reference future releases (including a public platform and automated codebase) that cannot be evaluated during the review process. No dataset samples are provided for examination. Essential information is missing regarding arXiv paper collection methodology, selection criteria, and dataset construction processes. The dataset sizes (100 for Abstract-single, 30 for abstract-multi and related-multi) appear limited for a comprehensive benchmark. Table 2 needs clearer definitions for "Cases" and resolution of inconsistencies with Gov Report and WCEP numbers. Basic metrics like "average length" require proper definitions (e.g., words, tokens), and output length statistics are missing. The paper needs to address split overlaps, distribution statistics other than mean, and the representation of "multi-modal" information (L363). The appendix sections require completion, including the specification of the "expert LLM."  The sampling procedures and initial memory setup need detailed documentation.
- Testing multiple LLMs would benefit the experimental evaluation of the proposed framework. The current single-model approach limits our understanding of the method's generalizability. Including various LLMs of different sizes and architectures would provide valuable insights into the method's robustness and scalability across different model capabilities.
- The metric selection is poor and requires justification. The exclusive use of ROUGE-L(F1) without including ROUGE-1/2/Lsum metrics needs explanation. Standard practice suggests reporting multiple ROUGE variants for comprehensive evaluation. This would facilitate better comparison with existing literature and provide a more complete performance assessment. Despite LLM-as-a-judge WinRate, there are no model-based semantic metrics.
- The evaluation lacks an analysis of the quality of the generated thoughts. The paper must address thought length distributions, the percentage of repetitive thoughts, and qualitative assessment of the generated content. Such analysis would help validate the effectiveness of the thought generation process and identify potential areas for improvement.
- The paper requires statistics regarding the external hierarchical memory structure. Information about tree depth, node count, and memory utilization patterns would help readers understand the practical implications and scalability of the proposed approach. These details are essential for assessing the method's efficiency in real-world applications.
- The efficiency analysis needs development. While the authors acknowledge that "forming thoughts can be a lengthy process," the paper lacks a quantitative analysis of computational requirements and resource utilization. Given the claims about minimal computational impact, concrete efficiency metrics would strengthen the paper's practical contribution.
- No human evaluation and error analysis.
- The experimental methodology needs to specify the number of runs conducted for the reported results. This information is necessary to assess the performance metrics' reliability and stability.
- No statistical significance tests were performed on the reported results.
- The Related Work section requires expansion, particularly regarding the generate-then-read paradigm (Section 4.3) and memory-enhanced language models. Including relevant works like GenRead [1] and MedGENIE [2] would better position the current work within the field's context and highlight its unique contributions.
- Poor reproducibility. No information about hardware specifications, experimental time requirements, decoding strategies, and metric hyperparameters (e.g., see [3]).
- The paper needs to provide information about dataset and code availability and licensing terms.

[1] Generate rather than Retrieve: Large Language Models are Strong Context Generators, Yu et al., ICLR 2023
[2} To Generate or to Retrieve? On the Effectiveness of Artificial Contexts for Medical Open-Domain Question Answering, Frisoni et al., ACL 2024
[3] Rogue Scores, Grusky, ACL 2023

Presentational issues:
- A proper definition of "Hierarchical RALMs" (L70) is needed
- "LLMS" instead of "LLMs" (L74)
- Re-introduced LLM acronym definition (L91)
- Figures and tables appear too much dislocated w.r.t. their references in the text (e.g., Figure 1 at the beginning of page 2 references the end of page 3, with another figure in between). Given the high dependency between the description of the motivating example and its illustration, that part of the paper appears really difficult to read. The same applies to Table 4.
- Figure 1(c) lacks clarity, with Ti represented as derived from chunks K_{n-1} and K_n; poor right padding; a real leading example would have facilitated understanding. In this sense, the example of thought generation in Table 1 is great but does not help build an overall vision of the entire pipeline below the proposed method. It would have been better to propose shorter examples but in a more complete manner.
- Section 2.2 and Section 2.3 are repeated
- Section 2.4 is highly schematic and redundant
- Missing space before Table 2 reference (L289)
- Dot before citation and extra space after thousand digits (L293)
- No best and second-best legend for bold and underlined scores in Table 3
- "with with" (L385)
- Fig/Table and its reference number should not be on distinct lines (L386). See also L400.
- Missing dot (L403)
- Inconsistent Figure and Section reference notation (e.g., "Fig." and "Fig", "Section" and "Sec")
- Inconsistent space between Figures bottom and caption start
- Unreferenced Tables (e.g., Table 5 and Table 6)
- Figure 5 requires additional context to be fully informative (e.g., the origin of thoughts)

**Questions:**

The reviewer has no specific questions to raise at this stage. The authors are encouraged to carefully consider each point detailed in the weaknesses section, as addressing these limitations would significantly strengthen the paper's contribution.

---

### Note · Authors · 2024-12-02

I have read and agree with the venue's withdrawal policy on behalf of myself and my co-authors.